# Nomogram Predicting the Risk of Postoperative Major Wound Complication in Soft Tissue Sarcoma of the Trunk and Extremities after Preoperative Radiotherapy

**DOI:** 10.3390/cancers14174096

**Published:** 2022-08-24

**Authors:** Zhengxiao Ouyang, Sally Trent, Catherine McCarthy, Thomas Cosker, Duncan Whitwell, Harriet Branford-White, Christopher Leonard Maxime Hardwicke Gibbons

**Affiliations:** 1Department of Orthopedics, The Second Xiangya Hospital, Central South University, Changsha 410011, China; 2Nuffield Orthopaedic Centre, Oxford University Hospital Foundation Trust, Oxford OX3 7LD, UK; 3Department of Oncology, Churchill Hospital, Oxford University Hospital Foundation Trust, Oxford OX3 7LE, UK

**Keywords:** soft tissue sarcoma, wound complication, preoperative radiotherapy, nomogram, limb preservation

## Abstract

**Simple Summary:**

Preoperative radiotherapy increases the risk of postoperative wound complication in the treatment of soft tissue sarcoma. This retrospective study evaluated risk factors and aimed to develop a nomogram for predicting major wound complication requiring secondary surgical intervention. We found that age, tumour size, and metastasis at presentation were independent risk factors of major wound complication. The nomogram constructed in the study effectively predicts and quantifies the risk of major wound complication.

**Abstract:**

Preoperative radiotherapy increases the risk of postoperative wound complication in the treatment of soft tissue sarcoma (STS). This study aims to develop a nomogram for predicting major wound complication (MaWC) after surgery. Using the Oxford University Hospital (OUH) database, a total of 126 STS patients treated with preoperative radiotherapy and surgical resection between 2007 and 2021 were retrospectively reviewed. MaWC was defined as a wound complication that required secondary surgical intervention. Univariate and multivariate regression analyses on the association between MaWC and risk factors were performed. A nomogram was formulated and the areas under the Receiver Operating Characteristic Curves (AUC) were adopted to measure the predictive value of MaWC. A decision curve analysis (DCA) determined the model with the best discriminative ability. The incidence of MaWC was 19%. Age, tumour size, diabetes mellitus and metastasis at presentation were associated with MaWC in the univariate analysis. Age, tumour size, and metastasis at presentation were independent risk factors in the multivariate analysis. The sensitivity and specificity of the predictive model is 0.90 and 0.76, respectively. The AUC value was 0.86. The nomogram constructed in the study effectively predicts the risk of MaWC after preoperative radiotherapy and surgery for STS patients.

## 1. Introduction

Radiotherapy (RT) combined with surgery can reduce the risk of recurrent disease in high-grade soft tissue sarcoma (STS) and is the standard recommended treatment [1,2]. With the timing of RT, preoperative RT has several potential advantages over postoperative RT in reducing long-term function impairment (fibrosis, joint stiffness, fracture) with lower radiation dose and field, the ability to evaluate tumour response, and without treatment delay or cancellation [3,4]. While preserving the maximal function of the limb with preoperative RT, postoperative acute wound complications occur in 9–35% of cases, which is much lower in postoperative RT [5], and remain a major concern in the management of STS patients [6,7].

With the improvement of orthopaedic wound care techniques, the management of most postoperative wounds no longer requires surgical intervention, which has facilitated the use of preoperative RT [5,8]. Wound complications requiring repeat surgery attributed to preoperative RT is a significant concern for surgeons and one that significantly affects patients’ quality of life. Thus, with an increasing demand for accurate and personalized risk assessment in major wound complication (MaWC), doctors require comprehensive and disease-specific knowledge. In this study, we aim to investigate the preoperative risk factors that relate to MaWC in patients with STS after preoperative RT, and to construct a nomogram to identify patients who are at a particularly high risk of MaWC and, ultimately, require reoperation and prolonged wound management.

## 2. Materials and Methods

A retrospective review of STS cases was carried out using the database of Oxford University Hospitals (OUH). After obtaining approval from our institutional review board and written informed consent from all patients, clinical, imaging and pathological data from 126 patients who underwent preoperative RT and resection of high-grade STS in the limb and trunk at Nuffield Orthopaedic Centre between 2007 and 2021 were collected. A total of 224 patients treated with postoperative RT, without surgery, or with retroperitoneal sarcoma were excluded. Positron emission tomography with fluorodeoxyglucose integrated with computed tomography (PET/CT) was used for disease staging preoperatively. All patients received preoperative RT with a total dose of 50 gray (Gy) in 25 daily fractions of 2 Gy each, five days a week. Surgery was performed between 3 and 6 weeks after the completion of preoperative RT. Following surgery, the patients were evaluated weekly until the wound was completely healed, then reviewed every 3 months for 2 years and every six months thereafter.

The data collected included patient gender, body mass index (BMI), smoking status, use of alcohol, mental status (depression and anxiety) and comorbidities (diabetes), as well as tumour site, size, volume, depth, histological subtype, maximum standardized uptake value (SUVmax) of PET/CT, metastasis at presentation and surgery type. Depression and anxiety were diagnosed by a GP (General Practitioner) before admission and recorded by nurses (the measurement of depression and anxiety self-assessment quiz is provided in the Appendix A). Smoking and alcohol consumption data were collected by trained nurses and roughly classified as smoker or non-smoker and alcohol drinker or non-drinker if the patients had any historical smoking or drinking records. The tumour size was determined by the measurement of the maximal cross-sectional diameter obtained on axial magnetic resonance imaging (MRI). The tumour site was subdivided into upper extremity, proximal lower extremity, distal lower extremity and trunk. The tumour depth was evaluated as deep or superficial to fascia. The tumour volume was measured on planning three-dimensioned imaging before RT. Wound complication such as haematoma, seroma, erythema, infection, wound dehiscence and lymphoedema that ultimately required secondary surgery necessitating general anaesthesia, drainage of hematoma, wound debridement, drainage of seroma, secondary wound closure or orthoplastic composite flap repair were considered MaWC.

The characteristics of patients were displayed as counts (percentage) for categorical variables and mean (standard deviation) for continuous variables, and the *p* values were derived using the chi-square test and t-test, respectively. A univariate and multivariate logistic regression analysis was applied to find the significant risk variables for MaWC. Variables with *p* value < 0.05 in the univariate analysis were included in the multivariable analysis. The diagnostic odds ratio (OR) and 95% confidence interval (CI) of each independent factor were calculated. All variables were included for building the nomogram for predicting MaWC. A bootstrapping approach was used for internal validation on the original study sample. A receiver operating characteristic (ROC) curve was applied to evaluate model discrimination and tested using bootstrap resampling (500 times). A calibration curve was employed to assess the calibration of the nomogram [9]. All analyses were performed using Empower (R) (http://www.empowerstats.com (accessed on 28 May 2022), X&Y solutions, Inc., Boston, MA, USA) and R (http://www.R-project.org, accessed on 21 August 2022) [9]. Statistical tests with *p* value < 0.05 were considered significant.

## 3. Results

Patients were followed postoperatively for an average of 71.82 months (range, 10–186 months). Among the 126 patients, 24 cases had MaWC after preoperative RT and surgery, and the incidence of MaWC was 19% (24/126). The mean age was 62 years with 68.5 in the MaWC group and 57.9 in the non-MaWC group. The mean BMI was 28.7 with 27.4 in the MaWC group and 28.9 in the non-MaWC group. The most common tumour site was the proximal lower limb in both groups. Most tumours were located deep in the tissue (80.2%) with 80.4% in the non-MaWC group and 79.2% in the MaWC group. With respect to the mean tumour volume and SUVmax, it appears that they were much higher in the MaWC group than in the non-MaWC group (710.1 cm^3^ vs. 434.6 cm^3^, 19.6 vs. 13.9); however, they showed non-significance after the univariate analysis. The tumour size in the MaWC group was larger than in the non-MaWC group (13 vs. 9.5 cm). The most common histology subtype was unclassified pleomorphic sarcoma in both groups. There were 8.7% patients with metastasis at presentation, and this proportion was higher in the MaWC group compared with the non-MaWC group (20.8% vs. 5.9%). Most patients experienced primary closure (92.9%). It seems that more patients in the MaWC group experienced plastic surgery closure compared with the non-MaWC group (16.7% vs. 4.9%), but without significance after the univariate analysis. There were four patients who experienced R1 resection, but no significant difference could be seen in the comparison between the two groups (4.2% vs. 2.9%). Table 1 summarizes the clinical and pathological characteristics of the patients.

After univariate and multivariate study, we found that age (OR: 1.04, 95%CI: 1.01–1.08, *p* = 0.009); diabetes (OR: 3.07, 95%CI: 1.01–9.71, *p* = 0.048); metastasis at presentation (OR: 4.21, 95%CI:1.17–15.22, *p* = 0.028); and tumour size (OR: 1.09, 95% CI: 1.01–1.17, *p* = 0.018) are risk factors in the univariate analysis. Age (OR: 1.08, 95%CI: 1.02–1.13, *p* = 0.004); metastasis at presentation (OR: 9.12, 95%CI: 1.21–68.67, *p* = 0.032); and tumour size (OR: 1.12, 95%CI: 1.01–1.24, *p* = 0.032) are independent risk factors of MaWC in the multivariate study. The variables for the univariate and multivariate study are summarized in Table 2. All the variates that could be evaluated preoperatively (age, gender, tumour site, SUVmax, metastasis at presentation, BMI, diabetes, smoking, alcohol, type of surgery, tumour size and tumour depth) were included in the predictive model and were incorporated into the nomogram. Due to the limited number of cases, when we set the tumour site according to four locations as we previously analysed, the nomogram could not be modelled. Thus, the tumour site was set as the lower limb vs. other location in nomogram 1, and proximal lower limb vs. other location in nomogram 2. As PETCT is not uniformly used worldwide and factors such as smoking, use of alcohol, anxiety and depression are not quantitatively defined, we built a simplified nomogram 3 without PETCT and these parameters.

The ROC curve for the predictive nomograms 1 and 2 are presented in Figure 1. The area under the ROC curve (AUC) and the corresponding 95% CI were estimated by bootstrap resampling (times = 500). It was 0.855 (95% CI: 0.770–0.917) with a sensitivity and specificity of 0.905 and 0.750 in nomogram 1 and 0.831 (95% CI: 0.742–0.898) with a sensitivity and specificity of 0.762 and 0.794 in nomogram 2, respectively. To further evaluate the discriminative ability and net benefits of the two models, a decision curve analysis (DCA) was performed. The DCA results of the two nomograms are shown in Figure 2. In general, nomogram 1 showed the highest net benefit. Therefore, nomogram 1 exhibited the best accuracy for risk prediction and the highest net benefit. Based on model 1 (Figure 3), the predictive model formula was as follows:Logit (MaWC) = −10.32344 − 0.75211 × gender + 0.07223 × age + 2.01095 × site + 0.04656 × SUVmax + 2.60662 × metastasis at presentation − 0.00630 × BMI + 0.88274 × smoking + 0.40016 × diabetes + 1.86011 × depression or anxiety − 0.16794 × alcohol + 1.37350 × type of surgery + 0.08594 × tumour size − 0.29359 × tumour depth(1)

The ROC curve for the predictive nomograms 3 are presented in Figure 4 and the area under the ROC curve (AUC) and the corresponding 95% CI were 0.822 (95% CI: 0.732–0.893) with a sensitivity and specificity of 0.917 and 0.637, respectively. Based on model 3 (Figure 5), the predictive model formula was as follows:Logit (MaWC) = −7.23569 − 0.19839 × gender + 0.04977 × age + 1.91307 × site + 2.27947 × metastasis at presentation − 0.01091 × BMI + 0.40016 × diabetes + 1.92471 × type of surgery + 0.07427 × tumour size − 0.35312 × tumour depth(2)

## 4. Discussion

Previous studies showed that RT combined with surgery could significantly reduce the rate of local recurrence in high-grade STS [4,5], meaning that the completion of the local treatment of STS is of utmost importance. In postoperative RT studies, it is reported that 15% of patients did not complete the combination treatment as planned due to MaWC [3]. A 12-week delay rate in receiving postoperative RT is 26% reported by Casabianca et al. [10] and 15% reported by Miller et al. [11]. In respect of oncological prognosis, preoperative RT would be preferable for the completion of local treatment without interruption from wound complication. However, the high risk of postoperative wound complication in preoperative RT compared with postoperative RT remains a substantial challenge [7,12,13]. With respect to the quality of life, function and oncologic outcomes, it is believed that MaWC needing a secondary procedure should be assessed separately. As with the development of integrated wound care, most wound complications have little effect on patients’ daily life. In this study, the rate of MaWC is 19%, which is consistent with the rate of 18% reported by Hui et al. [14], and is slightly higher than reported by Rosenberg et al. [15] and O’Sullivan et al. [5]. In these studies, the rate of secondary surgery considered as “a true clinically significant major complication” is 16%. The difference might be due to the different study group, as we only included patients with a high-grade tumour located in the trunk and limb, and they only included limb STS of both high and low grade. Studies also reported the rate of 11% of the secondary operation, in which only patients with a tumour located in the upper limb and who received a lower RT volume were included. Thus, the rate of 19% might be more accurate, according to the guideline which recommended RT application in high-grade sarcoma [1,2].

Given the complex and multifactorial nature of wound complications, recent studies examined whether specific clinical predictors could better identify patients at the greatest risk. Rosenberg et al. reported female gender, radiation outside their institution and low grade as independent risk factors for a secondary operation [15]. In this study, age, diabetes, tumour size and metastasis at presentation are risk factors for MaWC in the univariate analysis. In the multivariate regression analysis, age, metastasis at presentation and tumour size are independent risk factors. It is widely accepted that larger tumours were associated with a higher rate of complications, but there was wide variation in how tumour size was defined, which resulted in a high degree of heterogeneity [6,16]. Thus, tumour volume that was calculated by an oncologist before RT could consider more information than tumour size was also analysed in this study. However, we did not find it a risk factor for MaWC and it was not included in the nomogram, as we tested and found that the nomogram including the tumour volume had a lower AUC compared with the current one. This might because of the limitation of case numbers and the wide range of tumour volume.

To avoid MaWC, studies also suggested the important effect of immediate flap reconstruction on the postoperative wound [16,17,18], while others found the difference with or without plastic surgery to be non-significant [15,19,20]. Consistent with the latter, we did not find that the type of surgery was a significant risk factor for MaWC. However, for patients with a high risk of MaWC, Moor et al. reported that a substitution of irradiated soft tissue for healthy and well-vascularized soft tissue would be reasonable, especially with a tumour located in the adductor compartment of the thigh [16].

Adam et al. reported that patients with tumours located in the lower extremity with vascular involvement should be considered for immediate vascularized tissue transfer [8], which confirms the important effect of tumour site on MaWC. We did not find that tumour site was a significant risk factor for MaWC. This might be largely due to the stratification method, in which we divided tumour site into four groups according to guidelines, and made some adjustments to the recent literature that considers the proximal lower limb to be the most important risk factor [3,8,16]. As it is difficult to construct a nomogram with such stratification, a two categories method was applied in this study. Both lower limb vs. other location and proximal lower limb vs. other location were applied in the nomogram, and a better AUC was observed in nomogram 1 with the classification of tumour location by lower limb vs. other location.

Metastasis at presentation is an independent risk factor for MaWC. Though it is recommended in guideline [1] that with metastatic disease, surgery and chemotherapy are first-line management, selected patients were treated with preoperative RT and surgery in this study. It is sometimes difficult to confirm whether the suspected lung or other site lesion is metastatic with staging and PETCT. Thus, for patients in which suspected metastatic disease cannot be excluded, caution should be made on MaWC.

With these results, we developed a prediction model to evaluate the MaWC risk in STS patients treated with preoperative RT and resection. Gender, age, smoking, alcohol, BMI, diabetes, depression and anxiety, tumour site, size and depth, SUVmax, metastasis at presentation and type of surgery, which were considered significant risk predictors in previous studies, were included in the model [6]. The AUC of the model is 0.855, indicating a good predictive value for MaWC. To our knowledge, this is the first study describing a nomogram for the prediction of MaWC based on preoperative factors.

This study had limitations. Some data (smoking and the use of alcohol, anxiety and depression) were not clearly and quantitatively defined, and PET-CT is not routinely used in the diagnostic work-up for all STS; thus, we developed a nomogram 3 without these parameters with lower AUC (0.822) for wider application. The data presented may be underpowered to detect a significant association of the variables with MaWC due to the potential for selection bias, particularly with respect to tumour type and for the indication of radiation. As noted previously, patients who underwent preoperative RT were those who were thought to be at a high risk for local recurrence. This was a single-centre multivariate prediction model development study, which may affect the accuracy and generalizability of the results. However, we used bootstrap resampling (time = 500) to estimate the AUC and the corresponding 95% CI to improve the accuracy of the model’s predictive value. In future studies, this established model should be applied in a multicentre study to validate its generalizability. The methodology in this study may be practical in clinical research and can be applied in different populations.

## 5. Conclusions

In conclusion, this study focuses on the impact of multiple personal variables and the synergistic interaction between variables that can increase the rate of postoperative MaWC. Age, tumour size and metastasis at presentation are independent risk factors. The nomogram showing a good predictive value for MaWC is an intuitive tool that provides clinicians with a graphical calculation of each predictor to assess the risk of MaWC. Clinicians can use this model easily and rapidly to clinically evaluate the risk of MaWC and administer an individualized strategy for STS patients.

## Figures and Tables

**Figure 1 cancers-14-04096-f001:**
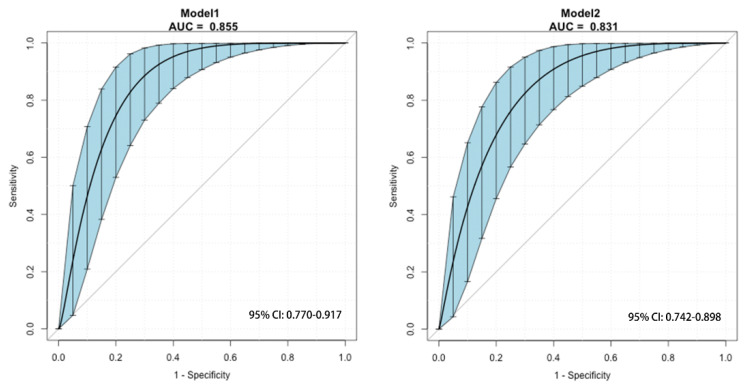
The ROC curve for the predictive model 1 and 2 using bootstrap resampling (times = 500). Shading shows the bootstrap estimated 95% CI with the AUC. ROC, receiver operating characteristic; AUC, area under the curve; CI, confidence interval.

**Figure 2 cancers-14-04096-f002:**
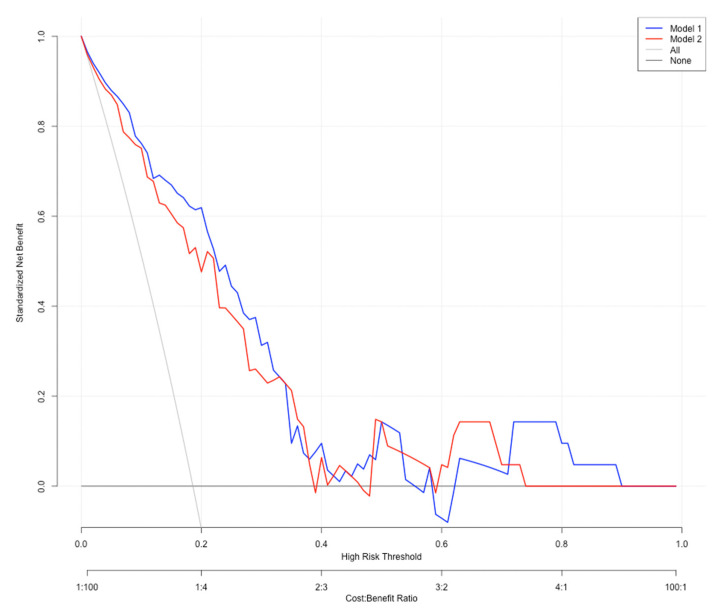
Decision curve analysis results of the nomograms. Net benefit curves of two predictive models. “None” line means net benefit when no participant is considered as having the outcome (major wound complication); “All” line means net benefit when all participants are considered as having the outcome. The preferred model is the model with the highest net benefit at any given threshold.

**Figure 3 cancers-14-04096-f003:**
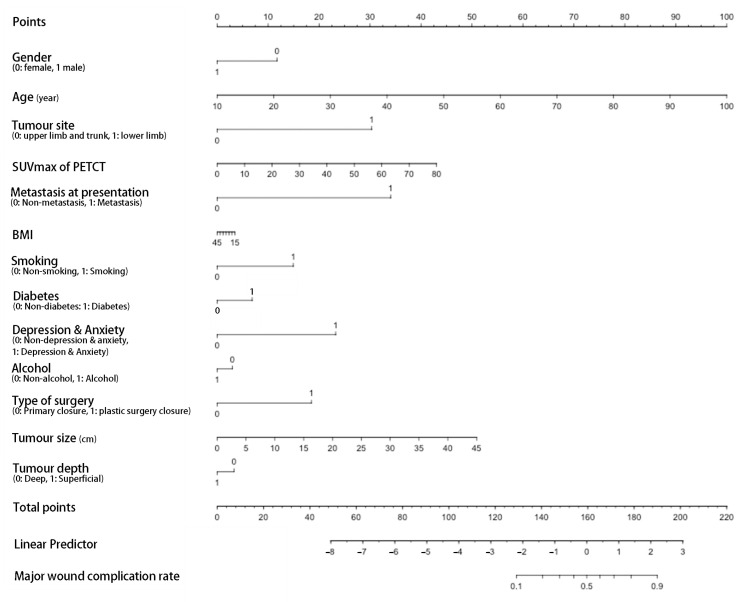
Nomogram 1 for prediction of postoperative major wound complication after preoperative radiotherapy and surgery. The point of each predictor could be assessed at the first line (Points) and the total points then could be calculated by summing up the points of each predictor and identified on the penultimate line. At last, the rate of MaWC could be assessed by the corresponding total points at the last line. BMI, body mass index.

**Figure 4 cancers-14-04096-f004:**
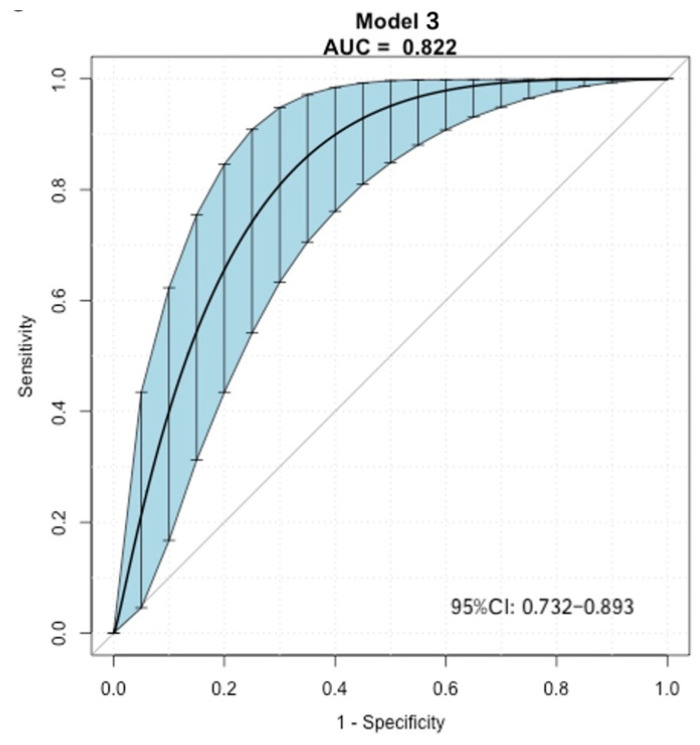
The ROC curve for the nomogram 3 using bootstrap resampling (times = 500). Shading shows the bootstrap estimated 95% CI with the AUC. ROC, receiver operating characteristic; AUC, area under the curve; CI, confidence interval.

**Figure 5 cancers-14-04096-f005:**
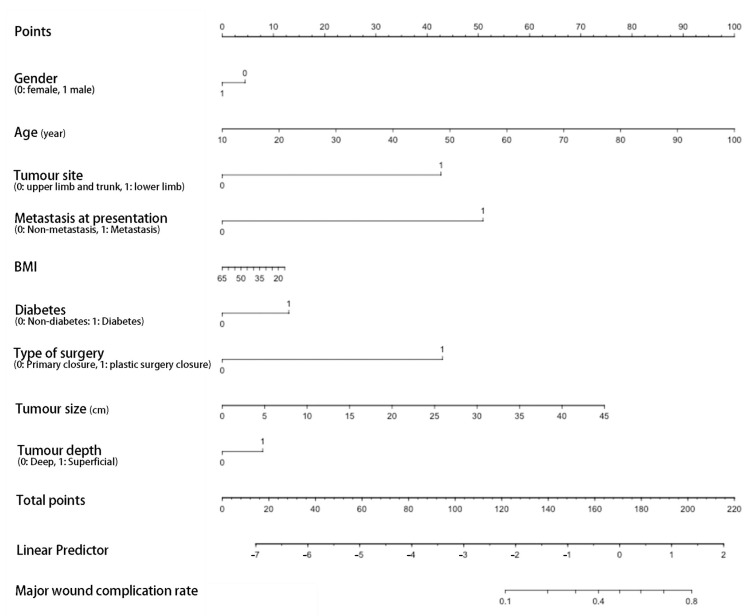
Nomogram 3 (without smoking, use of alcohol, anxiety and depression, SUVmax of PET-CT) for prediction of postoperative major wound complication after preoperative radiotherapy and surgery.

**Table 1 cancers-14-04096-t001:** Baseline characteristic of patients.

Category	Total, n	Non-MaWC n, %	MaWC, n, %	*p*-Value
Number of patients	126	102(81.0%)	24 (19.0%)	
Gender				0.741
Female	51 (40.5%)	42 (41.2%)	9 (37.5%)	
Male	75 (59.5%)	60 (58.8%)	15 (62.5%)	
Mean age (year)	62.0	57.9 ± 17.1	68.5 ± 15.4	0.009
Mean BMI	28.7	28.9 ± 7.2	27.4 ± 5.9	0.361
Diabetes				0.048
Non-diabetes	110 (87.3%)	92 (90.2%)	18 (75.0%)	
Diabetes	16 (12.7%)	10 (9.8%)	6 (25.0%)	
Smoking				0.127
Non-smoking	80 (63.5%)	68(66.7%)	12 (50.0%)	
Smoking	46 (36.5%)	34 (33.3%)	12 (50.0%)	
Alcohol				0.179
Non-alcohol	43 (34.1%)	32 (31.4%)	11 (45.8%)	
Alcohol	83 (65.9%)	70 (68.6%)	13 (54.2%)	
Depression or anxiety				0.467
Non-depression or anxiety	115 (91.3%)	94 (92.2%)	21 (87.5%)	
Depression or anxiety	11 (8.7%)	8 (7.8%)	3 (12.5%)	
Tumour site				0.181
Upper limb	11 (8.7%)	11 (10.8%)	0 (0.0%)	
Proximal lower limb	75 (59.5%)	60 (58.8%)	15 (62.5%)	
Distal lower limb	19 (15.1%)	13 (12.8%)	6 (25.0%)	
Trunk	21 (16.7%)	18(17.6%)	3 (12.5%)	
Tumour depth				0.892
Deep	101 (80.2%)	82 (80.4%)	19 (79.2%)	
Superficial	25 (19.8%)	20 (19.6%)	5 (20.8%)	
Mean tumour volume (mean ± SD, cm^3^)	434.6	376.1 ± 173.7	710.1 ± 240.2	0.065
Mean tumour size (mean ± SD, cm)	10.2 ± 6.0	9.5 ± 5.3	13.0 ± 7.9	0.018
PETCT SUVmax (Mean ± SD)	15.0 ±13.8	13.9 ± 9.7	19.6 ± 16.7	0.090
Histology type				0.686
Myxoid liposarcoma	23 (18.3%)	19 (18.6%)	4 (16.7%)	
Other liposarcoma	8 (6.4%)	5 (4.9%)	3 (12.5%)	
Myxoidfibrosarcoma	19 (15.1%)	16 (15.7%)	3 (12.5%)	
Synovial sarcoma	11 (8.7%)	10 (9.8%)	1 (4.2%)	
Undifferentiated pleomorphic sarcoma	13 (10.3%)	11 (10.8%)	2 (8.3%)	
Leiomyosarcoma	12 (9.5%)	11 (10.8%)	1 (4.2%)	
Unclassified pleomorphic sarcoma	25 (19.8%)	20 (19.6%)	5 (20.8%)	
Unclassified spindle-cell sarcoma	6 (4.8%)	4 (3.9%)	2 (8.3%)	
Others	9 (7.1%)	6 (5.9%)	3 (12.5%)	
Metastasis at presentation				0.028
Non-metastasis	115 (91.3%)	96 (94.1%)	19 (79.2%)	
Metastasis	11 (8.7%)	6 (5.9%)	5 (20.8%)	
Type of surgery				0.066
Primary closure	117 (92.9%)	97 (95.1%)	20 (83.3%)	
Plastic surgery closure	9 (7.1%)	5 (4.9%)	4 (16.7%)	
Surgery margin				0.758
R0	122 (96.8%)	99 (97.1%)	23 (95.8%)	
R1	4 (3.2%)	3 (2.9%)	1 (4.2%)	

Others including rhabdomyosarcoma, malignant peripheral nerve sheath tumour, angiosarcoma.

**Table 2 cancers-14-04096-t002:** Univariate and multivariate analysis of significant predictors of wound complications.

Risk Factor	Univariate Analysis			Multivariable Analysis		
	OR	95%CI	*p*-Value	OR	95%CI	*p*-Value
Age	1.04	1.01–1.08	0.009	1.08	1.02–1.13	0.004
Diabetes	3.07	1.01–9.71	0.048	2.46	0.57–10.42	0.226
Metastasis at presentation	4.21	1.17–15.22	0.028	9.12	1.21–68.67	0.032
Tumour size (cm)	1.09	1.01–1.17	0.018	1.12	1.01–1.24	0.032

## Data Availability

The data presented in this study are available on request from the corresponding author. The data are not publicly available due to privacy restrictions.

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
