# Peer review of "Nomogram Predicting the Risk of Postoperative Major Wound Complication in Soft Tissue Sarcoma of the Trunk and Extremities after Preoperative Radiotherapy"

_cancers, 2022, doi:10.3390/cancers14174096_

Round 1

Reviewer 1 Report

The authors present their significant effort in developing a nomogram for the prediction of major postoperative wound complication in soft tissue sarcoma after preoperative radiotherapy. It concerns a very interesting and important issue in the treatment of soft tissue sarcoma, which needs validation. The manuscript is well written. Some comments are to be made.

Firstly, I would suggest to add to the title that it concerns soft tissue sarcoma of the trunk and extremities (not head and neck, intra-abdominal or retroperitoneal). The authors do not mention how anxiety and depression are (objectively) measured and diagnosed. ‘Smoking’ and ‘alcohol’ should also be defined. Is sporadically smoking and use of alcohol included? Moreover, worldwide PET-CT is not routinely used in the diagnostic work-up for soft tissue sarcoma. Hence inclusion of SUV max at PET-CT created a problem for the use of the nomogram.

Since the above parameters are not objectively defined or not used in every patient, the presented nomogram cannot be routinely utilized. Therefore I would suggest development of a nomogram without ‘anxiety’ and SUV max at PET-CT and if ‘smoking’ and ‘alcohol’ can not be clearly defined, also without these parameters.

Author Response

Comments: The authors present their significant effort in developing a nomogram for the prediction of major postoperative wound complication in soft tissue sarcoma after preoperative radiotherapy. It concerns a very interesting and important issue in the treatment of soft tissue sarcoma, which needs validation. The manuscript is well written. Some comments are to be made.

Response: We are grateful for the valuable comments. The comments from the reviewer were perspective and helpful in revision and improvement of the manuscript. We have revised the manuscript according to the comments provided.

Comments: Firstly, I would suggest to add to the title that it concerns soft tissue sarcoma of the trunk and extremities (not head and neck, intra-abdominal or retroperitoneal).

Response: We thank the reviewer for the thoughtful comments. We have added the trunk and extremities to the title.

Comments: The authors do not mention how anxiety and depression are (objectively) measured and diagnosed. ‘Smoking’ and ‘alcohol’ should also be defined. Is sporadically smoking and use of alcohol included? Moreover, worldwide PET-CT is not routinely used in the diagnostic work-up for soft tissue sarcoma. Hence inclusion of SUV max at PET-CT created a problem for the use of the nomogram.

Since the above parameters are not objectively defined or not used in every patient, the presented nomogram cannot be routinely utilized. Therefore I would suggest development of a nomogram without ‘anxiety’ and SUV max at PET-CT and if ‘smoking’ and ‘alcohol’ can not be clearly defined, also without these parameters.

Response: We thank the reviewer for the thoughtful comments. The retrieved data from the note of electronic medical records of NHS patients, mental disorder such as anxiety and depression are evaluated by GP before admission to the Nuffield Orthopaedic Centre and recorded by nursing staffs. A depression and anxiety self-assessment quiz (https://www.nhs.uk/mental-health/self-help/guides-tools-and-activities/depression-anxiety-self-assessment-quiz/) is used to help measure and diagnose anxiety and depression. As ‘Smoking’ and ‘alcohol’ are difficult to quantify according to the records, we defined such varieties as a categorical variable as shown in the tables and nomogram figure: 0 means non-smoking and non-alcohol, and 1 means smoking and alcohol. Cases with sporadic smoking and useof alcohol are included in 1 group. We have also added explanation of in Materials and Methods (line 74-80) and limitation in the Discussion (236-239).

We agree that PET-CT is not widely used world wide for staging in STS treatment but standard practice in the UK. However, it is reported that PET-CT at diagnosis provides a very useful predictive tool for patients with STS, thus we wonder the role of SUVmax of PET-CT in prediction of wound complication. We agree with the suggestion of the reviewer that develop of a nomogram 3 without these parameters that could not be evaluate accurately and conveniently. The sensitivity and specificity of the nomogram 3 is 0.917 and 0.637, respectively (AUC: 0.822, 95%CI: 0.732-0.893). Based on the decreased AUC of the nomogram 3 comparing with nomogram 1 and 2, we put the figures and results in the supplementary file for anyone preferring to use the simplified model.

Reviewer 2 Report

Thank you for the opportunity to review this article, focusing on risk factors for wound complications following neoadjuvant radiotherapy in the treatment of sarcomas.

The topic is always very relevant. The study is well conducted and the article well written. In particular, I congratulate the authors for the depth of the analysis performed. Furthermore, the proposal of a nomogram is interesting for its potential clinical utility.

The introduction is brief but comprehensive, the methodology well presented, the results are clear, the discussion covers the key points, and the conclusions are consistent with the results. Although no groundbreaking data emerge from this study, the paucity of evidence in the literature on the topic makes the article worthy of publication in my opinion. I only have two considerations for the authors 

1) the "many advantages" of pre-operative radiotherapy are mentioned in the introduction; I think it would be useful to elaborate more on the respective advantages and disadvantages between pre- and post-operative radiotherapy, for the benefit of readers who are less confident about the matter;

2) it would be appropriate to clarify in the methods what kind of univariate statistical analyses were conducted.

Thank you.

Author Response

Comments: The topic is always very relevant. The study is well conducted and the article well written. In particular, I congratulate the authors for the depth of the analysis performed. Furthermore, the proposal of a nomogram is interesting for its potential clinical utility.

The introduction is brief but comprehensive, the methodology well presented, the results are clear, the discussion covers the key points, and the conclusions are consistent with the results. Although no groundbreaking data emerge from this study, the paucity of evidence in the literature on the topic makes the article worthy of publication in my opinion. I only have two considerations for the authors 

Response: We thank the reviewer for the valuable comments. The comments from the reviewer were highly insightful and constructive for our revision and significant improvement of our manuscript. We have checked carefully and revised the manuscript according to the comments.

Comments: The "many advantages" of pre-operative radiotherapy are mentioned in the introduction; I think it would be useful to elaborate more on the respective advantages and disadvantages between pre- and post-operative radiotherapy, for the benefit of readers who are less confident about the matter;

Response: We thank the reviewer for the thoughtful comments. We have revised the introduction. (line 44-49)

Comments: it would be appropriate to clarify in the methods what kind of univariate statistical analyses were conducted.

Response: We thank the reviewer for the valuable comments. We have revised the methods and added: The characteristics of patients were displayed as counts (percentage) for categorical variables and mean (standard deviation) for continuous variables, and the p value were derived using the chi-square test and t-test, respectively. In line 89-91.

Round 2

Reviewer 1 Report

The authors present a nomogram for prediction of postoperative wound complications after radiotherapy for soft tissue sarcoma of the extremities and the trunk. Firstly, I would like to congratulate the authors for their impressive effort and improvements according to the comments that were made. The responses to the previous comments were more than adequate. However, I would like to make some minor comments.

I would suggest adding to the measurement of depression and anxiety by the GP that a depression and anxiety self-assessment quiz (https://www.nhs.uk/mental-health/self-help/guides-tools-and-activities/depression-anxiety-self-assessment-quiz/) is used, as noted in the response to the reviewer’s comments.

Finally, thanking the authors for agreeing that PET-CT is not uniformly used worldwide in the diagnostic work-up for soft tissue sarcomas and providing a nomogram without the PET-CT parameter, I would strongly recommend providing this additional nomogram (#3) in the main text and not in a supplementary file. Many readers may unfortunately not make the effort to open a supplementary file.

Author Response

Response: We are grateful for the valuable comments. The depression and anxiety self-assessment quiz forms have been uploaded in supplementary file and explained in Materials and Methods (line 79-80).

Nomogram 3 has been provided in the main text and we revised the results part (Page 7) and limitation.
